# Toll-like Receptor 2 Is Associated with the Immune Response, Apoptosis, and Angiogenesis in the Mammary Glands of Dairy Cows with Clinical Mastitis

**DOI:** 10.3390/ijms231810717

**Published:** 2022-09-14

**Authors:** Xu Bai, Xueying Wang, Ting Lin, Weitao Dong, Yuan Gao, Peng Ji, Yong Zhang, Xingxu Zhao, Quanwei Zhang

**Affiliations:** 1College of Life Science and Technology, Gansu Agriculture University, Lanzhou 730070, China; 2Gansu Key Laboratory of Animal Reproductive Physiology and Reproductive Regulation, Lanzhou 730070, China; 3College of Veterinary Medicine, Gansu Agriculture University, Lanzhou 730070, China

**Keywords:** clinical mastitis, bacteria, mammary gland, TLR2, CASP8, Tie2

## Abstract

Toll-like receptor 2 (TLR2) plays a crucial role in bacterial recognition and the host immune response during infection. However, its function and downstream biological processes (BPs) in the mammary glands (MGs) of Holstein cows with clinical mastitis (CM) are not fully understood. This study aimed to comprehensively identify the BPs and differentially expressed proteins (DEPs) associated with the bacterial response and TLR2 using data-independent acquisition (DIA) proteomic data. A possible mechanism for the action of TLR2 was proposed, and the results suggested that the expression levels of TLR2 and caspase 8 (CASP8) were positively correlated with the apoptosis of MGs. The expression patterns of TLR2 and TEK receptor tyrosine kinase 2 (Tie2) were negatively correlated with angiogenesis. These results indicated that TLR2 might promote apoptosis in mammary epithelial cells (MECs) and vascular endothelial cells (VECs) via upregulation of CASP8 expression, and inhibition of angiogenesis in VECs via downregulation of Tie2 expression in dairy cows with CM. In conclusion, TLR2 is associated with inflammation, apoptosis, and angiogenesis in the MGs of dairy cows with bacteria-induced mastitis. These results contribute to a deeper understanding of the pathogenic mechanisms and provide the knowledge needed for developing the prevention and treatment of dairy mastitis.

## 1. Introduction

Cow mastitis, an inflammatory disease of the mammary gland (MG), is caused mainly by bacterial intra-mammary infection [1]. It is the most prevalent disease in cows worldwide. In recent decades, researchers have made significant efforts to prevent and treat dairy cow mastitis from different perspectives [2,3,4]. However, clinical mastitis (CM) is still prevalent, with an average incidence of 5% in dairy cows. Researchers have thus far been unable to identify biomarkers from milk, serum, somatic cells, and MGs that can be used for diagnostic and therapeutic purposes.

The most common and prevalent udder pathogens are *Escherichia coli* (*E. coli*), *Staphylococci aureus* (*S. aureus*), and *Streptococcus dysgalactiae*, as isolated from mastitis-infected milk [5]. MGs are invaded by these pathogens, leading to the activation of host immune regulation and defense responses, host anti-infection responses, and regulation of virulence gene expression [6,7]. The optimal strategy for preventing and treating of pathogen-induced mastitis is to enhance host immunity and eliminate pathogenic bacteria in the host. Due to the increasing drug resistance of pathogenic bacteria and limited drug availability, there is an urgent need to identify alternative, safe, and cost-effective bactericides and biomarkers for the treatment of dairy mastitis.

To date, an increasing number of omics technologies have been applied to understand better the biological adaptations occurring during bovine mastitis and to investigate changes in the expression of genes and proteins associated with mastitis. For example, proteomics data have shown that cathelicidins and serum amyloid A are involved in inflammation, chemotaxis of immune cells, and antimicrobial defense and can be used as biomarkers for improved diagnosis of mastitis [8,9,10]. The upregulated expression of collagen type I alpha 1 (COL1A1) and inter-alpha (globulin) inhibitor H4 (ITIH4) in mastitis may be associated with host tissue damage and repair during the late stages of infection [10]. An increasing number of biomarkers have been identified from different stages and subtypes of dairy mastitis, most of which are associated with the immune system and inflammatory response. These biological processes play a crucial role in mastitis and other perinatal diseases in animals. In addition, studies have suggested that various pro-inflammatory proteins secreted by either mammary epithelial cells (MECs) or immune cells participate in bacterial defense [11]. Thus, it is necessary to understand the initial identification and immunosuppressive and antibacterial response processes of bacteria in mastitis. Previous studies have also suggested that pathogens penetrate the physical barrier of the udder canal, and the host innate immune system detects and responds to bacteria through pattern recognition receptors (PRRs), particularly toll-like receptors (TLRs) [12,13]. TLRs are the major sensors stimulated during bacterial infection that initiate the inflammatory process by sensing PRRs [14,15]. However, the functions and activation mechanisms of PRRs, including TLRs, are not completely understood, particularly in dairy mastitis.

The present study aimed to identify the biological processes (BPs) and differentially expressed proteins (DEPs) associated with the bacterial response and TLRs in cows with CM according to the data-independent acquisition (DIA) proteomic data. In addition, we aimed to further analyze the possible mechanisms underlying these processes and validate them using a variety of molecular biological assays. These findings may improve our understanding of the biological processes of bacterial response and TLRs during bovine mastitis and aid the development of prevention and control programs and treatment protocols for dairy cows with CM.

## 2. Results

### 2.1. Identification of Candidate GO Terms and DEPs Associated with Bacterial Response

We focused on the candidate DEPs from Gene Ontology (GO) associated with bacterial response. A total of 13 GO terms (Appendix A) were selected from the BP terms in the DIA data (Figure 1A). After overlapping the repeated DEPs, 45 DEPs (Appendix A) were identified in the 13 BP terms. The heat map showed that the expression levels of these DEPs were relatively consistent in the triplicates of the Con group. However, significantly differential expression patterns were exhibited compared to the CM group (Figure 1B). The volcano plot showed that the 45 DEPs consisted of two downregulated and 43 upregulated proteins (Figure 1C). The Venn diagram suggested that 12 BP terms were associated with bacterial response and shared only one DEP (Toll-like receptor 2, TLR2), except for the BP term associated with the antibacterial humoral response (Figure 1D). The results indicated that these 45 DEPs play important roles in the bacterial response and CM, especially TLR2.

### 2.2. Identification of the Pathways and DEPs Interacting with TLR2

From the significantly different pathways, 10 pathways (Appendix A) and 134 interacting DEPs (Appendix A) were selected (Figure 2A). Among these, six pathways were associated with infectious diseases, whereas two pathways were involved in immune diseases. The heat map showed that these 134 DEPs were differentially expressed between the two groups. The expression levels of most of these DEPs in triplicates of the Con group were relatively consistent. However, the expression levels of these DEPs in triplicates of the CM group were significantly differentially expressed (Figure 2B). The volcano plot suggested that the 134 DEPs consisted of 120 upregulated and 14 downregulated proteins (Figure 2C). Taking TLR2 as the interacting core, a protein–protein interaction (PPI) network of the 134 DEPs was constructed, and the results showed that the 133 DEPs directly interacted with TLR2 (Figure 2D). Only one node was joined between the 77 DEPs and TLR2. In contrast, two or more nodes were connected between the 56 DEPs and TLR2, especially STAT1, PTK2B, and CYBA. These results indicated that the function of the TLR2 and selected DEPs are associated with infection and immunity in MGs, especially those infected by bacteria.

### 2.3. Molecular Mechanism of TLR2 in the MGs of CM Cows

We analyzed the structure of TLR2, and the results suggested that TLR2 consists of three regions, including the extracellular (EC) region enriched with β-folds, transmembrane (TM) region, and intracellular (IC) region enriched with α-helices (Figure 3A). The EC region is composed of ten leucine-rich repeats (LRRs) responsible for protein adhesion and pathogen identification. The TM region is responsible for performing cellular localization. The IC region is composed of an intracellular Toll-interleukin 1-resistance (TIR) region responsible for signal transduction between TLR2 and downstream proteins or kinases. According to the results, the potential mechanism of TLR2 in the MGs of dairy cows with bacteria-induced CM was deduced (Figure 3B). Exogenous factors such as peptidoglycan (PGN), lipopolysaccharides (LPS) of gram-positive bacteria, and pathogen-associated molecular patterns (PAMPs) can be identified and bound by the EC region of TLR2. Transduction into the intracellular (IC) region might trigger different biological processes by binding to different targets. For instance, TLR2 could directly or indirectly promote inflammation and tissue injury and suppress immune reactions by regulating the expression of some immune factors, such as interleukins (ILs) and nuclear factor kappa B (NF-kB). TLR2 promotes apoptosis by regulating caspase 8 (CASP8) expression through the myeloid differentiation primary response 88 (MyD88)-dependent pathway. TLR2 can also regulate angiogenesis or inflammatory cell infiltration by binding to activator protein 1 (AP-1) or interacting with TEK receptor tyrosine kinase 2 (Tie2) in the blood vessels.

### 2.4. Distribution and Expression Patterns of TLR2 in the MGs of Cows

Hematoxylin and eosin (H&E) staining showed that the morphological structures of the alveoli and blood vessels were complete and clear in the Con group (Figure 4(A1–D1)). In contrast, the morphological structures of alveoli and blood vessels in the CM group were incomplete and infiltrated with massive inflammatory cells, including neutrophils (NEUT) and lymphocytes (LY) in the lumens and around the alveoli and blood vessels (Figure 4(A2–D2)). Immunohistochemistry (IHC) results showed that the positive TLR2 signals were distributed in the cytoplasm of MECs in the alveoli (Figure 4(A3,A4)), vascular endothelial cells (VECs) of blood vessels (Figure 4(B3–D3,B4–D4)), and vascular smooth muscle cells (VSMCs) of arterioles and venules (Figure 4(B3,C3,B4,C4)) in the two groups. The immunopositive signals of TLR2 in the CM group were stronger than in the Con group. Positive TLR2 signals were also present in the cytoplasm of macrophages (Mø), NEUTs, and LYs (Figure 4(E1–E4)). The nuclear conformation of polymorphonuclear cells exhibited two or three segmented nuclei in the alveoli. Immunopositivity for the TLR2 protein was absent in the negative control groups (Figure 4(A5–D5,A6–D6)). Compared with the Con group, the expression level of *TLR2* mRNA was significantly upregulated in the CM group (*p* < 0.05). The expression level of TLR2 was relatively consistent in triplicate in the Con group. Nevertheless, it was significantly differentially expressed in triplicate in the CM group (Figure 4F). The TLR2 protein was presented in the MGs of the two groups (Figure 4G). Compared to the Con group, the relative expression level of the TLR2 protein was significantly upregulated in the CM group (*p* < 0.05), which was in accordance with the *TLR2* mRNA levels in the Con group and the triplicates. The results of the expression pattern analysis of *TLR2* mRNA and protein in the two groups were consistent with those from the DIA proteomic data.

### 2.5. Co-Localization Analysis of TLR2 in the MGs of Cows

The immunofluorescence (IF)-positive signals of the TLR2 and CK18 proteins were co-localized in the cytoplasm of the MECs in the two groups (Figure 5A). Based on the IF signals of DAPI, TLR2, and CK18, we found that the nuclei and MECs were regularly arranged in the MGs of the Con group. However, they were diffusely distributed in the MGs in the CM group (Appendix A). In addition, TLR2 proteins were also present in MA cells, such as NEUTs, in the CM groups. The labeled alveoli in group Con were intact and neatly arranged, without non-specific expression of TLR2 and CK18 in the acinus cavities, while the labeled alveoli in the CM group were distributed diffusely; moreover, large amounts of MECs were shed in the acinus cavities. The IF-positive signals of CD31, α-SMA, and TLR2 proteins were co-distributed in the cytoplasm of the VECs and VSMCs in the arterioles and venules of the MGs of the two groups (Figure 5B,C). The IF positive signals of CD31 and TLR2 proteins were co-located in the cytoplasm of the VECs in the capillaries of the MGs of the two groups. The positive TLR2 signals were distributed in the cytoplasm of the VECs and VSMCs with a regular morphology in the Con group, but were scattered in the CM group (Appendix A). Co-localization analysis showed that the VECs and VSMCs in blood vessels were stained with different degrees of TLR2, CK18, CD31, and α-SMA. The morphological structures of the arterioles and venules showed an intact blood vessel wall embedded with smooth muscles in group Con. However, the blood vessel walls of the arterioles, venules, and capillaries were damaged to different degrees, particularly the IF signals of TLR2, which exhibited obvious changes in the CM group. Strong IF signals of TLR2 were also exhibited in the arterioles of the two groups. Moreover, shed VECs and VSMCs were present in the vascular lumen or perivascular regions in the CM group.

### 2.6. Validation of Tie2 and CASP8 in the MGs of Cows

Co-localization analysis showed that the TLR2, CK18, and CASP8 proteins were co-located in the MECs of the MGs, and apoptotic bodies were present in the MECs of the CM group (Figure 6A). The IF-positive signals of CASP8 in the CM group were stronger than those in the Con group (Appendix A). TLR2, CD31, and CASP8 proteins were co-located in the VECs of blood vessels in the MGs. Apoptotic bodies were observed in the VECs of the CM group (Figure 6B). The IF-positive signals of CASP8 in the VECs of the CM group were also significantly stronger than those in the Con group (Appendix A). TLR2, CD31, and Tie2 proteins were co-located in the VECs of blood vessels in the MGs (Figure 6C). The IF-positive signals of Tie2 in the VECs of the CM group were weaker than those in the Con group (Appendix A). Apoptosis of MECs and VECs in the MGs was assessed using DAPI staining. The results showed a large number of nuclei with chromatin condensation in the MECs of alveoli and VECs of blood vessels in the CM group, indicating that cell apoptosis was present in the MGs (Figure 6D,E and Appendix A). Compared to the Con group, the apoptosis rate was 5.64-fold in the alveoli of the CM group and 8.35-fold in the blood vessels of the CM group (Figure 6F,G and Appendix A). Compared with the Con group, the expression level of *Tie2* mRNA was significantly downregulated in the CM group, and the expression level of *CASP8* mRNA was significantly upregulated in the CM group (*p* < 0.05) (Figure 6H,I). Western blotting showed that Tie and CASP8 proteins were present in the MGs of the two groups (Figure 6J). Compared with the Con group, the relative expression level of Tie2 was significantly downregulated in the CM group, and the relative expression level of CASP8 was significantly upregulated in the CM group (*p* < 0.05), which was in accordance with the results of the mRNA and DIA proteomic data.

## 3. Discussion

A wide variety of bacteria and their products can trigger different kinds of inflammation that cause cytokine release, host cell apoptosis, and immune reactivity [16], which are also the main causes of dairy mastitis. In recent decades, several studies have focused on the function and mechanism of inflammation-related cells in the immune system and inflammation induced by pathogens in mastitis [6]. However, candidate targets and their regulatory mechanisms in the MGs of Holstein cows with CM have not been well characterized. Therefore, understanding the infection mechanism and pathogenic and signal transduction mechanisms in the host cell of the pathogens is essential for developing prevention and control programs and treatment protocols in dairy cows with CM.

### 3.1. Immune Responses Are an Essential Step against Bacterial Infection in the MGs of Dairy Cows

Infection by intra-mammary bacteria can activate bacterial responses and the MG immune system in different ways [17]. MGs perform various immunological functions, including recognizing and discriminating between foreign invading agents and molecules produced by the organism that produce protective effects against infectious agents [18]. In the present study, we focused on the DEPs associated with the bacterial response in the MGs of Holstein cows with CM according to DIA proteomic data. Results from the analysis of the GO selection identified 13 BP terms and 45 DEPs in the MGs (Figure 1). These BP terms are related to innate immunity, immunological recognition, and signal transduction between pathogens and host cells, similarly to the DEPs. Bacteria have different cell wall structures, including those primarily composed of LPS, PGN, and lipoteichoic acid (LTA), which are recognized by specific plasma membrane receptors and constitute pathogen-associated molecular patterns (PAMPs) [18,19]. For instance, some DEPs, especially TLR2 and CD14, can induce an initial immune response via various immuno-modulators [20,21].

### 3.2. TLR2 Plays a Crucial Role in the Immune Response of MGs

Among these DEPs, we found that TLR2 was a crucial target amongst multiple BP terms. Previous studies have also shown that the formation of the TLR2 heterodimer is the initial step leading to significant innate immune responses, development of adaptive immunity to pathogens, and protection from immune sequelae related to infection with pathogens [22]. In addition, some DEPs, such as ANXA3 [23], ALDOA [24]**,** and PGLYRP3 [25], are associated with immune responses in different diseases. However, the functions and mechanisms of these DEPs remain unknown and require further study. Next, we focused on the pathways involving TLR2. The results suggest that most of these pathways are associated with infectious or immune diseases (Figure 2). The PPI network results also proved that these candidate DEPs directly interacted with TLR2. The expression patterns of the DEPs selected for analysis of BP terms and pathways were displayed with relative consistency in the individuals of the Con group, whereas they were significantly differentially expressed. This may be due to the infection dose and difference in the growth rate of the intrusive pathogens, which are reflected in the differences in the immune response induced by these organisms [17].

### 3.3. Multiple Functions of TLR2 Cause Different Biological Processes in MGs

Based on the results of bioinformatics analysis, the potential regulatory mechanisms of TLR2 in the MG were characterized (Figure 3), and the results suggested that the PAMPs of pathogens might combine with LRRs of TLR2, triggering different biological processes by binding with different targets. Insertions within LRRs are crucial for PAMP recognition in TLRs signal transduction processes [26]. The role of TLR2 in immunity and inflammation has already been well established in related cells [19,22,27]. H&E and IHC results also confirmed that TLR2 was located in the inflammatory cells (Figure 4), indicating that TLR2 may be associated with immunity and inflammation. Notably, TLR2 was distributed in the alveoli and blood vessels, particularly in the MGs of the CM group, indicating that TLR2 plays an important role in the alveoli and blood vessels. The expression patterns of *TLR2* mRNA and proteins were positively correlated with CM. Alveoli, especially MECs, have been widely studied for their role in the synthesis and secretion of inflammatory chemokines and pro-inflammatory cytokines, which are crucial for the prevention and treatment of mastitis [28]. On the other hand, it has also been illustrated that vascular diseases or abnormal angiogenesis often involve a complex interplay between cells intrinsic to the vascular wall (VECs and VSMCs) and invading immune cells [29].

### 3.4. TLR2 Is Associated with Apoptosis and Angiogenesis in the MGs of Dairy Cows

The results of the colocalization analysis revealed that TLR2 was colocalized in MECs, VECs, and VSMCs (Figure 5). Compared to the Con group, diffuse distribution of alveoli shed MECs in acinus cavities, and damaged blood vessel walls were the noticeable morphological changes in the MGs of the CM group. Intact barriers formed by epithelial cells in the alveoli and endothelial cells in the blood vessels are essential for homeostasis and immunity in the MGs of dairy animals [30]. Changes in the permeability of alveoli and blood vessels [31], which lead to the leakage of blood components into milk (defined as hemorrhagic mastitis) or clot formation, prevent complete milk removal and cause CM in the MGs. We hypothesized that these changes in the barriers might be induced by the apoptosis of MECs in the alveoli and VECs in the blood vessels of MGs. This hypothesis was confirmed by colocalization and apoptosis rate analyses (Figure 6A). The expression levels of *TLR2* and *CASP8* mRNA and proteins in the MGs of the CM group were significantly upregulated compared to those in the Con group. These results suggest that TLR2 and CASP8 participate in the apoptosis of MECs and VECs in the MGs. Previous studies have demonstrated that the upregulation of TLR2 can activate CASP8, resulting in the activation of downstream effector caspases and the onset of apoptosis in MECs [32] and VECs [33]. Moreover, we found that TLR2 and Tie2 were co-located in the VECs of blood vessels in the MGs. The blood vessel walls in the CM group were incomplete, which might be due to the downregulation of *Tie2* mRNA and protein levels in the MGs of the CM group. Decreasing Tie2 can increase the permeability of the blood vessel walls [34]. The ANG/Tie system is the second endothelial cell-specific ligand-receptor signaling pathway necessary for vascular remodeling and permeability changes [35,36].

### 3.5. Potential Limitations

The small sample size (*n* = 3) used in this study leads to a lack of significance for some variables, which may affect test performance and accuracy. However, strict diagnostic criteria for sample collection should ensure good usability power of the data (88.5 to 100%) [37]. Furthermore, future studies should aim to elucidate the regulatory mechanisms of TLR2 in apoptosis, angiogenesis, and inflammation at a cellular level in MGs from Holstein cows.

### 3.6. Clinical Implications of the Findings

Collectively, our results revealed that TLR2 is not only associated with immunity and inflammation but also plays crucial roles in apoptosis and angiogenesis in the MGs of Holstein cows with or without CM. These findings can help to understand the pathogenesis of CM in dairy animals and further provide references for methods to improve the diagnosis and treatment of CM. 

## 4. Materials and Methods

### 4.1. Study Design and Sample Collection

All Holstein cows were undergoing their 3rd to 5th lactation and were 60–250 days in milk, and were enrolled from a commercial farm (WuZhong City, Ningxia Hui Autonomous Region, China). The cows had normal metabolism and were not affected by other diseases. Veterinary udder examination was performed to assess criteria such as redness, hotness, swelling, and painful sensation, as described previously [38]. The MGs of the cows with typical clinical symptoms were classified into CM groups. Fresh milk samples from healthy lactating Holstein cows and CM (*n* = 8 per group) cows were collected under aseptic conditions for somatic cell count (SCC) and pathogen isolation and identification (Appendix A) to confirm the clinical diagnosis results. The healthy Holstein (SCC < 1 × 10^5^ cells/mL, control, Con group, *n* = 3) cows without pathogen and the CM cows induced using Staphylococcus aureus (SCC ≥ 13 × 10^5^ cells/mL, clinical mastitis, CM group, *n* = 3) were selected in this study and then delivered to the slaughterhouse. Fresh MGs from the two groups were collected and stored in liquid nitrogen or fixed with 4% paraformaldehyde immediately after slaughter, as previously described [28]. According to the DIA proteomic sequence data, the DEPs included in the enriched GO terms were associated with bacterial responses, and the candidate DEPs from the pathways encompassing the TLR2 were selected for further study. The functions of TLR2 were validated using different assays. This study was approved by the Animal Ethics Committee of Gansu Agriculture University, Lanzhou, China (no. GSAU-AEW-2018-0128).

### 4.2. Bioinformatics Analysis

DIA proteomic sequence data (accession number IPX0003382000/PXD028100) from the ProteomeXchange database [28] were used for Gene Ontology (GO) and Kyoto Encyclopedia of Genes and Genomes (KEGG) pathway annotation using R packages. The GO terms and KEGG pathways with *Q* value ≤ 0.05 were considered statistically significant. In the present study, we focused on the DEPs included in the GO terms associated with bacterial responses from significant biological processes (BP, *p* < 0.05, and *Q* < 0.05). According to the results of GO function analysis, the candidate DEPs from the pathways encompassing the TLR2 protein and associated with bacteria and bacterial response (*p* < 0.05, *Q* < 0.05) were selected for pathway analysis. The images, including heat maps, circular graphs, bubble diagrams, volcano plots, and Venn diagrams of the DEPs, were drawn using R and the OmicShare online platform (https://www.omicshare.com/tools/) [39,40]. Protein and protein interaction (PPI) networks of the candidate DEPs, GO terms, and pathways were constructed using STRING v. 10.0 (https://cn.string-db.org/) [41] and Cytoscape 2.8.1 [42], including ClueGO and Ingenuity pathway analysis (IPA) [43]. The structural domain and function of TLR2 were predicted as described previously [44,45]. The structure and signal transduction diagram of TLR2 was constructed using Adobe Illustrator 2020 (Adobe Systems, San Jose, CA, USA).

### 4.3. Hematoxylin and Eosin (H&E) and Immunohistochemical (IHC) Staining

Fixed tissues were embedded in paraffin (Solarbio, Beijing, China) and cut into 5 μm-thick sections using a microtome (Leica, Wetzlar, Germany). H&E and IHC staining assays were carried out as described previously [28,46]. The immunopositivity reaction of the TLR2 protein was performed according to the manufacturer’s instructions using the ABC staining system (Bioss, Beijing, China). A rabbit polyclonal antibody (Bioss, Beijing, China) was incubated overnight at 4 °C. Images were captured using a microscope (Olympus, Tokyo, Japan). All staining assays were performed in triplicates.

### 4.4. Immunofluorescence (IF) Staining

IF staining assays were performed as previously described [28,46]. The sections were labelled with primer antibodies at different dilution rates (Appendix A). The sections were incubated with the appropriate antibodies (CY3 for α-SMA and CK18, FITC for TLR2, and CY5 for CD31; Bioss, China) after incubation with the primary antibodies. The nuclei were labeled using 4′, 6-diamidino-2-phenylindole (DAPI). Fluorescence signals and images were captured using a Panoramic DESK slice-scanner system (3D HISTECH Co., Budapest, Hungary). All immunostaining assays were performed in triplicate.

### 4.5. DAPI Staining for Cell Apoptosis

Apoptosis of MECs and vascular endothelial cells (VECs) in the MGs of the Con and CM groups was observed using DAPI staining, as described previously [47]. The Con group was used as the control, and the apoptotic rates of MECs and VECs were calculated as described previously [47,48].

### 4.6. RNA Isolation, cDNA Synthesis, and qPCR Assays

Total RNA was extracted from the MGs of the two groups and was used for cDNA synthesis. RNA isolation, cDNA synthesis, and qRT-PCR assays were performed as described previously [28]. qRT-PCR primers (Appendix A) were designed using Premier software (5.0) and were synthesized by Qinke Biotech Co. Ltd. (Yanglin, Shanxi, China). The relative expression levels of *TLR2*, *CASP8*, *Tie2*, and *GAPDH* mRNA in the MGs were detected using qRT-PCR. The relative expression level of *GAPDH* was used as an endogenous control, and the expression levels of *TLR2* and *GAPDH* in the Con group were used as controls. Results were calculated using the 2^−ΔΔCT^ method [28]. All PCR assays were performed in triplicates.

### 4.7. Western Blot

The relative expression levels of TLR2, CASP8, Tie2, and β-actin proteins in the MGs of the Con and CM groups were examined by Western blotting. Total protein was extracted from 100 mg of each sample using RIPA (Solarbio, Beijing, China). The procedures were performed as previously described [28,46]. The primary antibodies were incubated with different dilution rates at 4 °C overnight. The optical densities of the bands were scanned and quantified using Image-Pro Plus 6.0 software (Media Cybernetics Co., Rockville, MD, USA). The expression level of β-actin was used as an endogenous control. The expression level of the TLR2 protein in group Con was used as the control. All immunoblot assays and optical densities were performed in triplicates.

### 4.8. Statistical Analysis

No assumptions for normality of data were made due to the small sample size. All variables were expressed as the median (range), and were analyzed using SPSS 22.0 software (SPSS Inc., Chicago, IL, USA) with the Wilcoxon rank sum procedure (α = 0.05). Significant differences in relative expression of mRNA, protein, and apoptosis related to inflammation or apoptosis or angiogenesis between the Con and CM were determined. Graphs were constructed using Prism 5.0 (GraphPad Software Inc., San Diego, CA, USA). A two-sided value of *p* < 0.05 was considered statistically significant.

## 5. Conclusions

A total of 13 BP terms and 45 DEPs associated with bacterial responses in the MGs of Holstein cows with CM were identified using bioinformatics methods according to the DIA data, especially TLR2. A total of ten pathways and 134 interacting DEPs associated with TLR2 were selected from the significantly different pathways. The potential regulatory mechanisms of TLR2 in MG were characterized, and the results showed that TLR2 was associated with the immune response, apoptosis, and angiogenesis in the MGs of dairy cows with CM. The verification results suggested that TLR2 was presented in the MECs, VECs, and VSMCs. Compared to the Con group, TLR2 expression levels were significantly upregulated in the CM group. Co-localization analysis showed that TLR2 was co-located and positively correlated with apoptosis via CASP8 in the MECs and VECs in the MGs. It was negatively correlated with angiogenesis via Tie2 in the VECs in the MGs. These results expanded our understanding of TLR2 functions in the MGs of Holstein cows with CM, and will aid in developing prevention, control, and treatment protocols in dairy cows with CM.

## Figures and Tables

**Figure 1 ijms-23-10717-f001:**
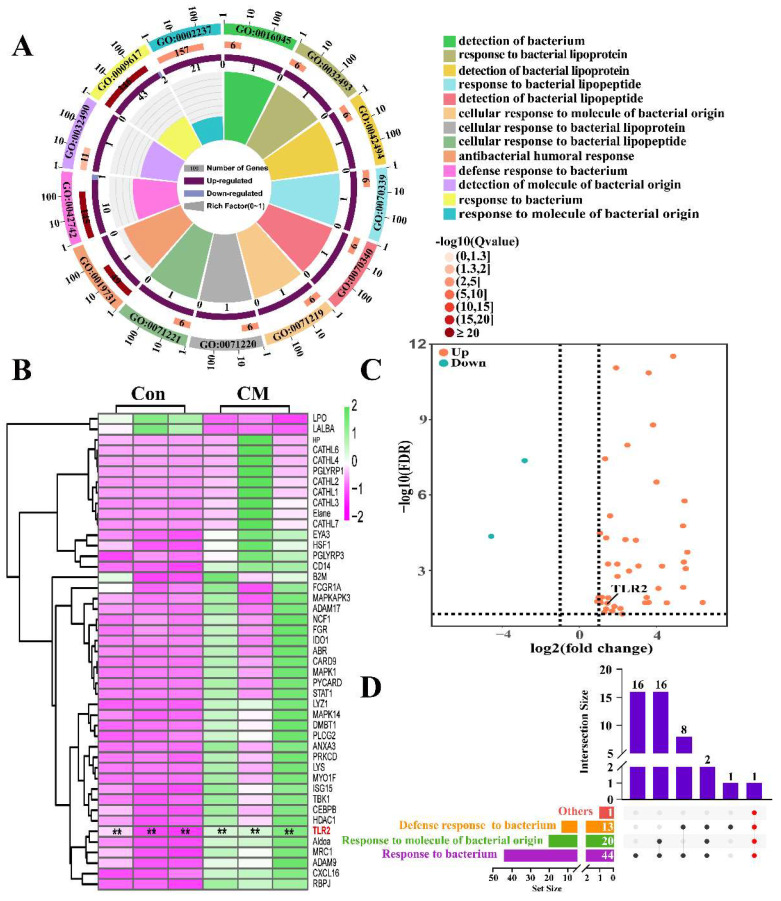
Identification of the candidate differentially expressed proteins (DEPs) associated with bacterial response according to the Gene Ontology terms. (**A**) The selected 13 biological process (BP) terms associated with bacterial response according to the data-independent acquisition (DIA) proteomic data. (**B**) Heat map of the 45 DEPs, included in the 13 BP terms associated with bacterial response. (**C**) Volcano plot of the 45 DEPs including two down- and 43 upregulated DEPs. (**D**) Upset diagram of the 13 BP terms and the 45 DEPs. DEPs, differentially expressed proteins; Con, control; CM, clinical mastitis. ** represents *p* < 0.01.

**Figure 2 ijms-23-10717-f002:**
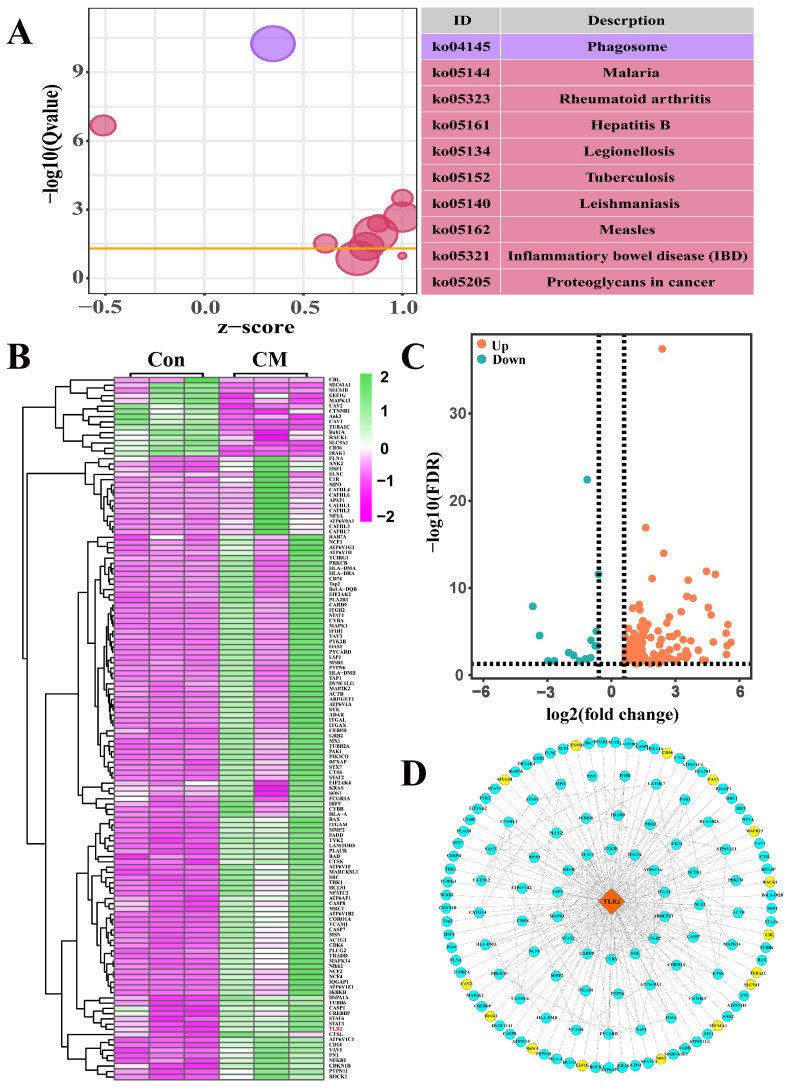
Identification of the pathways and differentially expressed proteins (DEPs) interacting with TLR2 according to the data-independent acquisition (DIA) proteomic data. (**A**) The selected pathways included TLR2 from the 68 significantly different pathways of the DIA proteomic data. (**B**) Heat map of the 134 DEPs included in the 10 pathways. (**C**) Volcano plot of the 134 DEPs, including 120 up- and 14 downregulated DEPs. (**D**) The constructed PPI network according to the 134 DEPs. The blue color represents upregulated DEPs. The yellow color represents downregulated DEPs. Con, Control; CM, Clinical mastitis.

**Figure 3 ijms-23-10717-f003:**
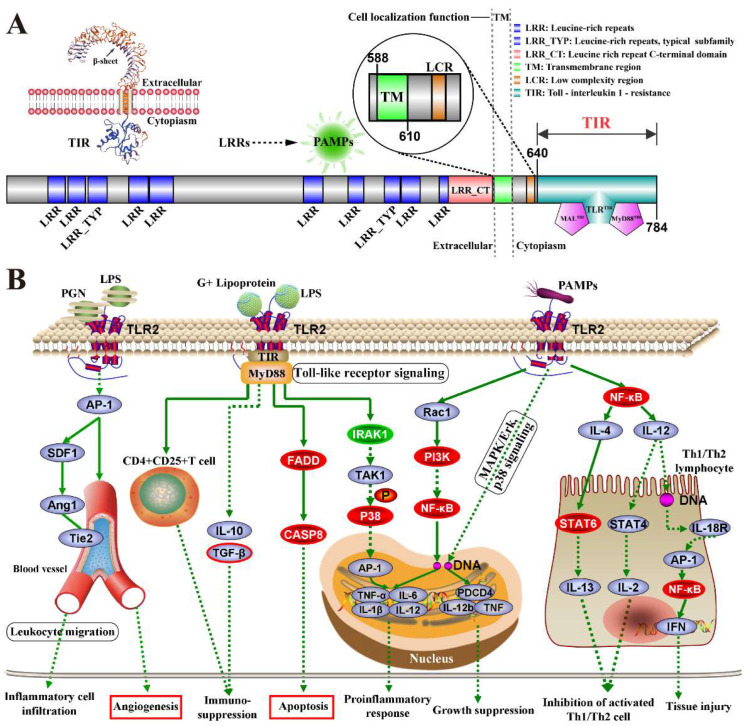
Deduced molecular mechanism of TLR2 in the mammary glands (MGs) of dairy cows with bacteria-induced clinical mastitis. (**A**) The predicted structure and function of TLR2. (**B**) The deduced molecular mechanism of TLR2 in the MGs according to the pathways. The red color represents the downregulated DEPs. The green color represents the upregulated DEPs.

**Figure 4 ijms-23-10717-f004:**
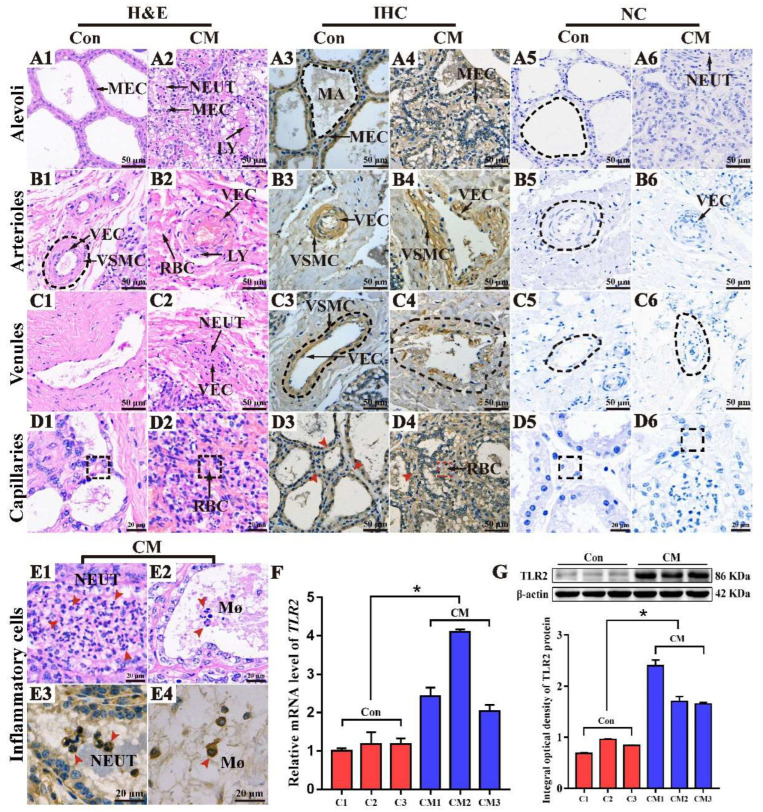
Distribution and expression patterns of TLR2 in the mammary glands (MGs) of Holstein cows. (**A1**–**D1**,**A2**–**D2**) Pathological variations of the alveoli (**A1**,**A2**), arterioles (**B1**,**B2**), venules (**C1**,**C2**), and capillaries (**D1**,**D2**) in the MGs of the two groups, respectively. (**A3**–**D3**,**A4**–**D4**) The subcellular location of the TLR2 protein in the alveolus, arteriole venules, and capillaries of the MGs in the two groups, respectively. (**A5**–**D5**,**A6**–**D6**) The negative control of the MGs, respectively. (**E**) Inflammatory cells of the CM group. (**E1**,**E2**) Morphology of NEUTs and macrophages (Mø). (**E3**,**E4**) The subcellular location of the TLR2 protein in NEUTs and Møs (1000×). (**F**,**G**) The relative expression of *TLR2* mRNA and protein in the Con and CM groups, respectively. MEC, mammary epithelial cells; MA, MG alveolus; VEC, vascular endothelial cell; VSMC, vascular smooth muscle cell; NEUT, neutrophil; VV, venous valve; RBC, red blood cell. Con, control group; CM, clinical mastitis group. Scale bar of 50 μm and 20 μm represents 400× and 800× magnification, respectively. * represents *p* < 0.05.

**Figure 5 ijms-23-10717-f005:**
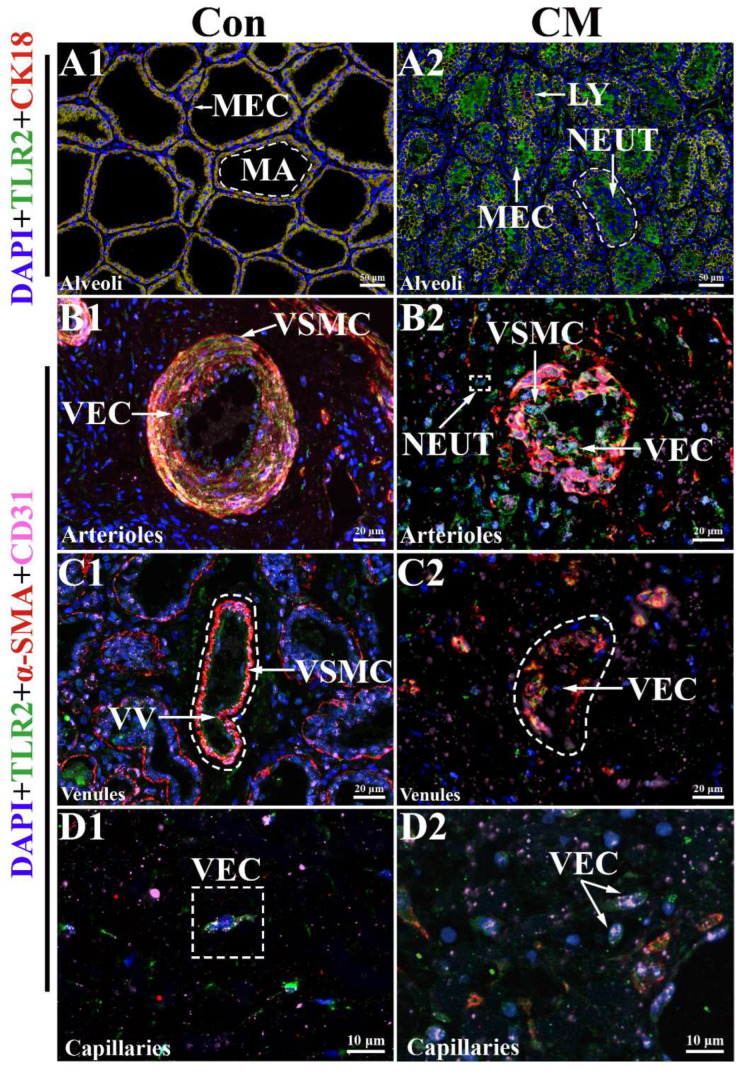
Co-localization analysis of TLR2 in the mammary glands (MGs) of Holstein cows. The nucleus of different types of cells labeled with DAPI. (**A**) Co-localization analysis of TLR2 and CK18 in the MGs of the two groups. (**B**–**D**) Co-localization analysis of TLR2, CD31, and α-SMA in the arterioles (**B1**,**B2**), venules (**C1**,**C2**), and capillaries (**D1**,**D2**) of the MGs in the two groups, respectively. MEC, mammary epithelial cells; MA, MG alveolus; VEC, vascular endothelial cell; VSMC, vascular smooth muscle cell; NEUT, neutrophil; VV, venous valve; LY, lymphocyte; Con, control group. CM, clinical mastitis group. Scale bar of 50 μm, 20 μm and 10 μm represents 400×, 630×, and 1000× magnification, respectively.

**Figure 6 ijms-23-10717-f006:**
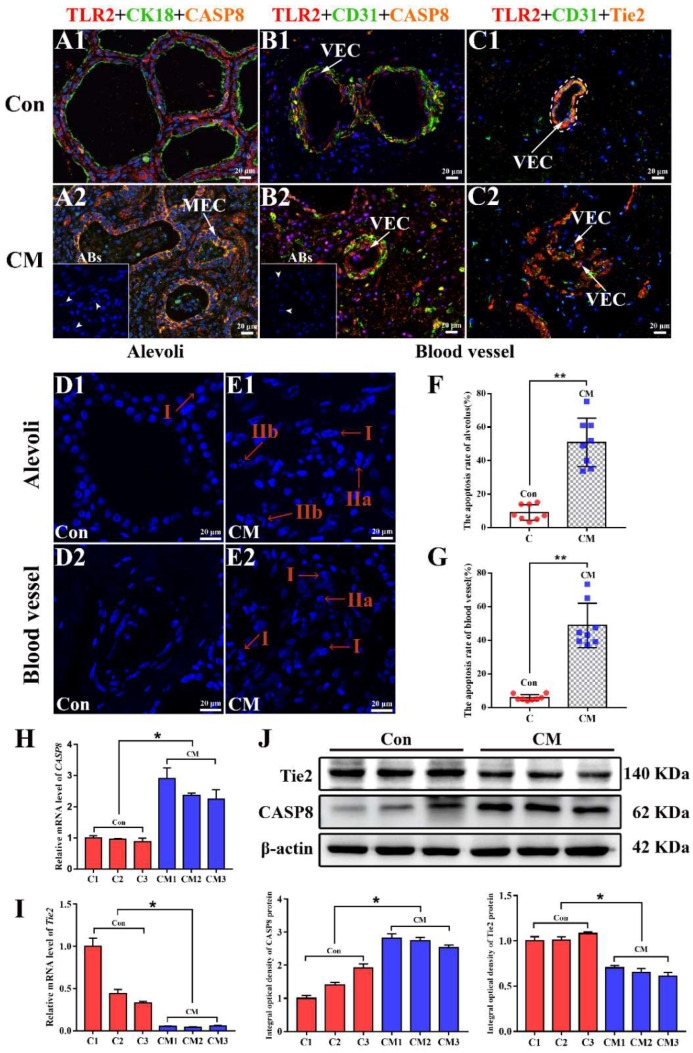
Validation of Tie2 and CASP8 in the MGs of Holstein cows. The nuclei of different types of cells labeled with DAPI. (**A**) Co-localization analysis of TLR2, CK18, and CASP8 in the mammary glands (MGs) of the two groups (400×). (**B**) Co-localization analysis of TLR2, CD31, and CASP8 in the two groups (400×). (**C**) Co-localization analysis of TLR2, CD31, and Tie2 in the two groups (400×). (**D**,**E**) The apoptosis of MECs (**D1**,**E1**) and VECs (**D2**,**E2**) in the alveoli and blood vessels of the two groups using DAPI staining (630×). (**F**) The apoptosis rate of MECs in the alveoli. (**G**) The apoptosis rate of VECs in the blood vessels. (**H**,**I**) The relative expression of *CASP8* and *Tie2* mRNA in the two groups, respectively. (**J**) The relative expression of CASP8 and Tie2 in the two groups. MEC, mammary epithelial cell; VEC, vascular endothelial cell; ABs, apoptotic bodies; Con, control group; CM, clinical mastitis group; Stage I, the nucleus is rippled or creased, with some chromatin in a concentrated state; Stage IIa, chromatin in the nucleus is highly condensed and marginalized; Stage IIb, the nucleus is cleaved into fragments and apoptotic bodies are produced. * represents *p* < 0.05 and ** represents *p* < 0.01.

## Data Availability

The data that support the findings of this study are available from the corresponding author upon reasonable request.

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
