# Peer review of "Toll-like Receptor 2 Is Associated with the Immune Response, Apoptosis, and Angiogenesis in the Mammary Glands of Dairy Cows with Clinical Mastitis"

_ijms, 2022, doi:10.3390/ijms231810717_

Round 1

Reviewer 1 Report

REVIEW

for the journal IJMS (ISSN 1422-0067)

Article “Toll-like receptor 2 is associated with the immune response, apoptosis, and angiogenesis in the mammary glands of dairy cows with clinical mastitis

Manuscript ID: ijms-1879593

Authors:  Xu Bai, Xueying Wang, Ting Lin, Weitao Dong, Yuan Gao, Peng Ji, Yong Zhang, Xingxu Zhao, Quanwei Zhang

1.In this study, the authors focused on differentially expressed proteins (DEPs) associated with bacterial detection and response using data-independent acquisition (DIA) proteomics. The results may contribute to a better understanding of the biological processes occurring during bovine mastitis.

2.I have questions about the homogeneity of the samples. “Fresh milk samples from lactating healthy Holstein cows and CM (n=8 per group) were collected under aseptic conditions from a commercial farm (WuZhong City, Ningxia Hui Autonomous Region, China) for pathogen isolation and identification. According to the results of diagnosis, healthy Holstein (control, Con/C group, n=3) and  CM cows (clinical mastitis, CM group, n=3) were selected and then delivered to the Slaughterhouse”.  My questions are: On what basis were these cows selected? What is the lactation and lactation period of cows, reproductive status, are metabolic or other diseases identified?

3.“CM cows (clinical mastitis, CM group, n=3)”.  Question: What pathogens were identified ?

4.Lines 338 - 339. The processes for udder examination of Holstein cows were performed as described previously [39]”. I think you should clarify.

5.I believe that in the methodology section, the authors should accurately describe the design of the study, taking into account the objectives of the study.

6.The statistical analysis section (lines 406 - 411) is very formal and does not describe what statistical methods and why were chosen to achieve the research objectives, given the research design. I suggest adding this part.

7.Lines 418 - 420. "These results deepen our understanding of the function of TLR2 in the MGs of Holstein cows with CM and will aid in the development of prevention, control, and treatment protocols in dairy cows with CM."  In my opinion, studies with larger homogeneous samples are needed to confirm this assertion.

8.The article is interesting, but the adjustments mentioned are recommended. Authors should pay attention to the research design, justification of sample selection and their homogeneity.

Sincerely, reviewer.

Reviewer 2 Report

Introduction

“Antibiotics and bactericides, such as enrofloxacin [and polyhexamethylene biguanide are the most effective drugs for this purpose and have been used for ……..”. This is a significant scientific error. Please delete.

The objectives of the study must be described clearly.

Procedures

The number of animals used in the study is small. The authors must justify this small number and also include a paragraph in the discussion about this limitation.

4.8. You used tools for data with normal distribution. Please present the evidence of normal distribution, otherwise please use non-parametric tests for the analysis.

Discussion

Please divide the discussion into sub-sections for better reading.

Also, please add a passage about the potential limitations of the study.

Also, please add a final paragraph about the clinical implication of the findings.

Final opinion: correction and re-evaluation.

Reviewer 3 Report

The manuscript reports on an investigation on Toll-like receptor 2 is associated with the immune response, apoptosis, and angiogenesis in the mammary glands of dairy cows with clinical mastitis. The idea is good considering that identifying new methods or tools to highlight the mechanism that triggers mastitis in cows is of considerable interest. The applied methods are appropriate and quite modern. The results are presented well, but there are too many interpretations of the data obtained. In the results, the data should be described without any kind of deduction.

The discussion is of adequate length and tries to justify the data obtained and sometimes with little sense.

The conclusions should be rewritten taking into account the real results obtained.

Round 2

Reviewer 2 Report

The manuscript only needs an extensive revision and improvement of language style before final acceptance.

Author Response

Comments and Suggestions for Authors

The manuscript only needs an extensive revision and improvement of language style before final acceptance.

Response: Thanks for your review. The language style was checked and revised by some native english speakers based on your suggestion. We believe that it is meet the standards.

Reviewer 3 Report

Many parts of the manuscript have been modified taking into account the suggestions of the reviewers.

Other parts such as the conclusions have not been changed and should be reviewed focusing on what has been achieved in the results.

Author Response

Comments and Suggestions for Authors

Many parts of the manuscript have been modified taking into account the suggestions of the reviewers.

Other parts such as the conclusions have not been changed and should be reviewed focusing on what has been achieved in the results.

Response: Thanks for your review. Actually, the conclusions have been revised. But it was not marked with colors in the manuscript. And now, we have revised it again based on your suggestion.